

# Suppression of the superfluid Kelvin-Helmholtz instability due to massive vortex cores, friction and confinement

Matteo Caldara[1], Andrea Richaud[2*], Massimo Capone[1,3] and Pietro Massignan[2]

**1** Scuola Internazionale Superiore di Studi Avanzati (SISSA),
Via Bonomea 265, I-34136, Trieste, Italy
**2** Departament de Física, Universitat Politècnica de Catalunya,
Campus Nord B4-B5, E-08034 Barcelona, Spain
**3** CNR-IOM Democritos, Via Bonomea 265, I-34136 Trieste, Italy

⋆ andrea.richaud@upc.edu

## Abstract

We characterize the dynamical instability responsible for the breakdown of regular rows and necklaces of quantized vortices that appear at the interface between two superfluids in relative motion. Making use of a generalized point-vortex model, we identify several mechanisms leading to the suppression of this instability. They include a non-zero mass of the vortex cores, dissipative processes resulting from the interaction between the vortices and the excitations of the superfluid, and the proximity of the vortex array to the sample boundaries. We show that massive vortex cores not only have a mitigating effect on the dynamical instability, but also change the associated scaling law and affect the direction along which it develops. The predictions of our massive and dissipative point-vortex model are eventually compared against recent experimental measurements of the maximum instability growth rate relevant to vortex necklaces in a cold-atom platform.

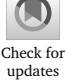

# 1   Introduction

The Kelvin-Helmholtz instability consists in the exponential amplification of infinitesimal fluctuations occurring at the interface between two fluid layers having a relative velocity $\Delta v$ [1,2]. This process leads to the breakdown of the laminar flow and to the onset of vortical structures [3–5]. This hydrodynamic instability, well known in the classical context and often considered a precursor of turbulence [6–9], is observed in various natural phenomena, including cloud formation [10,11], mixing of oceanic currents [12,13], and even astrophysical systems [14], where it is suspected to be a trigger mechanism for pulsar glitches [15]. The growth rate of this instability is $\sigma_c(q) = q\Delta v/2$ [5,16], $q$ being the wavenumber of the perturbation. In a realistic description, the two uniform flows are connected by a thin shear-layer that smooths out the velocity profile at the interface. It also introduces an ultra-violet cut-off $q^*$ which halts the runaway behaviour for small wavelengths (i.e. large $q$) of the instability growth rate [5]. In real classical fluids with viscosity $\mu > 0$, one has $q^* \propto \Delta v/\mu$ and the maximum growth rate scales as $\sigma_c^* = \sigma_c(q^*) \propto \Delta v^2$.

At low temperatures, some fluids display the remarkable property of superfluidity, i.e. they can flow with zero viscosity and dissipation. Superfluidity has been observed for example in liquid Helium-II [17], atomic Bose-Einstein condensates (BECs) [18], degenerate Fermi gases [19], and quantum fluids of light [20]. The macroscopic quantum behavior of all these fluids comes with several key distinctions with respect to their ordinary counterparts [21], such as the fact that vorticity exists only in the form of discrete filaments with quantized circulation.

A natural question is whether and to what extent the Kelvin-Helmholtz instability, explored and characterized rather exhaustively in classical fluids [5], has an analogue in superfluids, as it is the case for several well-known instabilities of ordinary fluids. For example, considerable attention has been devoted to the formation of von Kármán vortex streets in superfluid flows past obstacles [22–24], to the presence of boundary layers around the surfaces of moving objects [25], and to the Rayleigh-Taylor instability at the interface between two immiscible BECs [26–29]. As regards the Kelvin–Helmholtz instability, its first study in a superfluid system dates back to twenty years ago [30], where it was observed at the interface between $^3\mathrm{He} - A$ and $^3\mathrm{He} - B$ (see also the more recent Ref. [31]). Since then, a few theoretical and experimental works have followed, investigating its occurrence at the interface between a superfluid and a normal fluid [32,33], or between two different superfluids [34–41].

Subsequently, the superfluid Kelvin-Helmholtz instability has been (numerically) demonstrated within a single-component superfluid [42], a setup devoid of the complications (e.g. buoyancy effects) related to multicomponent systems. The ingenious protocol that was proposed is based on a progressive reduction of a potential barrier separating two channels, which leads to the merging of two counterflowing portions of the same superfluid, and hence to the seeding of an array of quantized vortices at the interface. Such an array was reported to

quickly break down as vortices precede to form increasingly larger clusters, mimicking the roll-up patches of vorticity, characteristic structures of the classical Kelvin-Helmholtz instability. Moreover, the instability growth rate $\sigma^*$ associated to the break down of this vortex array displays the same quadratic scaling $\sigma^* \propto \Delta v^2$ [43] as its classical counterpart. The same configuration of a quantized vortex array at the interface between two counter-propagating superflows in a two-dimensional (2D) BEC has been recently studied more in detail in Ref. [44]. Combining Gross-Pitaevskii simulations with a Bogoliubov approach, their thorough numerical analysis showed the occurrence of instabilities of different nature depending on the flow-velocity regimes. The quantized version of the hydrodynamic Kelvin-Helmholtz instability pops out at moderate velocities, while it washes out at supersonic flow velocities where other mechanisms emerge due to the coupling to acoustic excitations.

Cold-atom platforms are ideal to shed light on the nature of instabilities in quantum fluids due to shear flow. Zwierlein's group at MIT showed the fragmentation of a rapidly rotating elongated BEC into an array of droplets, as a consequence of a sheared velocity profile in the rotating frame [45]. More recently, Roati's group at LENS was able to characterize with unprecedented detail the Kelvin-Helmholtz instability in superfluid $^6$Li confined within an annular geometry [46].

Yet, many open questions remain about the detailed physical processes governing the breakdown of regular vortex arrays. In Bose superfluids, the presence of kelvon bound states localized at the vortex core (also at zero temperature) is responsible for a non-zero vortex mass which influences the vortex motion [47–50]. Similarly, in fermionic superfluids the motion of quantized vortices is affected by the interactions with the gas of elementary excitations and by the presence of Andreev bound states localized at the vortex cores [51–55]. This is the case, for instance, of the LENS experiment [46], where an instability growth rate $\sim 3$ times smaller than the one predicted by current theoretical models has been measured. They consider subsonic velocity differences where the Kelvin-Helmholtz instability is the dominant mechanism and its suppression cannot be ascribed to any coupling with acoustic excitations [44]. Given that, the observed mismatch has not received a theoretical explanation yet, hence it calls for further studies.

In this work, motivated by many state-of-the-art experimental facilities providing direct access to superfluid configurations with arbitrary geometries, we delve into the breakdown of regular rows and necklaces of quantized vortices due to the Kelvin-Helmholtz instability. On the one hand, we unify well-established results [43, 56, 57] relevant to point vortices in ideal fluids. On the other hand, we generalize them to test the robustness of the instability against the introduction of two classical ingredients that, while being often overlooked, are indeed present in most real superfluid systems. They are massive vortex cores and mutual friction arising from dissipative effects within the superfluid. In the framework of a suitably generalized point-vortex model, we carry out a detailed stability analysis of the shear layer present at the interface between counterflowing superfluids, demonstrating that massive vortex cores and dissipative processes, together with the proximity with the boundaries of the sample, are responsible for a partial or complete suppression of the superfluid Kelvin-Helmholtz instability. We also show that the presence of a finite core mass is responsible for a change of the asymptotic scaling of the instability growth rate, from quadratic to linear, i.e. $\sigma^* \propto \Delta v$. Whenever possible, our analysis is carried out in a fully analytical way, so to highlight the contribution that each of the aforementioned mechanisms has on the stabilization of quantized vortex arrays.

The structure of the manuscript is the following: in Sec. 2, we focus on the impact of a non-zero vortex-core inertial mass on the properties of regular vortex rows and show its rather general stabilizing effect. This analysis is motivated by the fact that, in many real superfluid systems, vortex cores are often filled, either accidentally or deliberately, by massive

particles that provide the topological excitations themselves with a non-zero inertial mass. Typical examples include tracer atoms in superfluid $^4$He [58,59], quasiparticle bound states both in fermionic [50,52–55] and bosonic superfluids [50] even at zero temperature, thermal atoms in atomic BECs [60], and atoms of a different species in two-component BECs [61–72]. Moreover, we introduce dissipation into our analysis to make our point-vortex model as accurate as possible. It turns out that dissipative processes indeed have a stabilizing effect on vortex arrays.

In Sec. 3, we rigorously incorporate the presence of an annular-like superfluid domain, a geometry which supports the prototypical realization of superflows with periodic boundary conditions [42,46]. The narrowness of the annulus further suppresses the necklace instability.

Sec. 4 is devoted to the analysis of massive vortex necklaces. The presence of core mass not only stabilizes the system, but also determines a change of the asymptotic dependence of the maximum instability growth rate on the relative velocity at the interface between the two counterflows.

In Sec. 5, we compare the predictions of our massive and dissipative point-vortex model with the results of a recent experimental characterization of the superfluid Kelvin-Helmholtz instability in a cold-atom platform [46]. This comparison offers insights into a potential estimate for the yet-undetermined transverse mutual-friction coefficient $\alpha'$, aiming at reconciling theoretical predictions with experimental observations. Finally, Sec. 6 is devoted to concluding remarks and future perspectives.

## 2 Vortex rows

When set into rotation, superfluids can host elementary excitations in the form of quantized vortices. The superfluid density is depleted in correspondence of the vortex cores, while the superfluid flow swirls around them. In quasi-two-dimensional (2D) configurations, the additional degrees of freedom associated to vortex-line bending become too high-lying in energy, and hence freeze out. When finite-compressibility effects can be neglected, and given their quantized vorticity, superfluid vortices can be effectively modeled as classical point-like particles (or as classical filaments in three-dimensional systems), simplifying the complex dynamics associated with their motion. This modeling approach proves particularly valuable when considering the interactions among multiple vortices. According to the principle of superposition of potential flows, the instantaneous velocity of each quantum vortex corresponds to the vector sum of the velocities induced by all other vortices within the system. In the present work we employ such a method since we deal with quasi-2D superfluids. A straight array of equally spaced vortices, the so called "vortex row" (schematically represented in Fig. 1) constitutes a stationary, but unstable, configuration. As is well known [43], in fact, any perturbation of this regular arrangement is amplified with a characteristic rate

$$\sigma_0 = \frac{\kappa q}{2a}\left(1 - \frac{qa}{2\pi}\right),\tag{1}$$

where $\kappa = h/m_a$ is the quantum of circulation, $m_a$ is the atomic mass of the superfluid, $a$ is the intervortex distance, and $q$ is the perturbation wavenumber. The maximum instability growth rate,

$$\sigma_0^* := \sigma_0(q^*) = \frac{\pi\kappa}{4a^2},\tag{2}$$

is the one associated to the most unstable mode $q^* = \pi/a$, which, in turn, corresponds to the minimum wavelength $(2a)$ that the lattice can support. An equivalent formulation of Eq. (2), $\sigma_0^* = \pi\Delta v^2/(4\kappa)$, is manifestly characterized by a *quadratic* dependence on the velocity difference $\Delta v = \kappa/a$ across the vortex row.

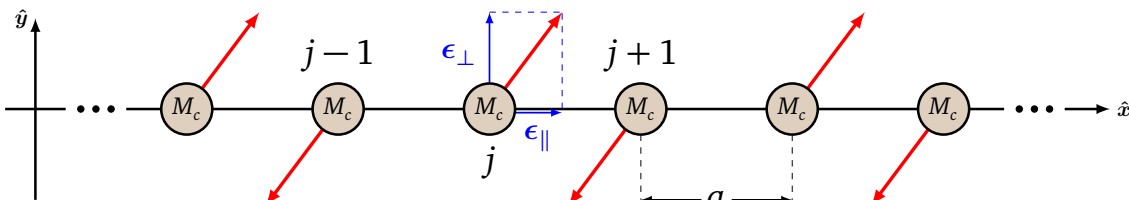

Figure 1: Schematic illustration of an infinitely extended vortex row featuring an intervortex distance $a$. Each vortex hosts a core mass $M_c$. The red arrows represent displacement vectors $\epsilon_j$, whose components are of the type $(-1)^j \left( \epsilon_\parallel, \epsilon_\perp \right)$.

## 2.1 Instability of a massive vortex row

As mentioned in the Introduction, in many real superfluid systems quantum vortices are often filled, either deliberately or accidentally, by massive particles, which provide them with an effective inertial mass. Interestingly, vortex mass can be an intrinsic property of a large class of quantum vortices, as it originates from kelvon modes or quasi-particle bound states localized in the vortex core [50]. The ensuing kinetic-energy term, often overlooked by well-established theoretical models [43, 56, 57], can be easily introduced within the Lagrangian description of superfluid vortex dynamics [64, 66]. To capture the impact of a finite core inertial mass on the superfluid Kelvin-Helmholtz instability, we start in this section by computing the maximum instability growth rate relevant to a massive vortex row. Later in Sec. 4, then, we will develop a similar computation for a massive vortex necklace inside a planar annulus.

We consider a massive vortex row inside a superfluid with atomic mass $m_a$ and 2D number density $n_a$. This configuration consists of a rectilinear regular chain of an infinite number of vortices. Each of them, labelled by the index $j \in \mathbb{Z}$, has the same quantum of circulation $\kappa$ and hosts a core mass $M_c$. The Lagrangian of the system reads

$$\mathcal{L}_{\text{plane}} = \sum_{j \in \mathbb{Z}} \left[ \frac{M_c}{2} \dot{\mathbf{r}}_j^2 + \frac{m_a n_a \kappa}{2} (\dot{\mathbf{r}}_j \times \mathbf{r}_j \cdot \hat{z}) + \sum_{i=1}^{+\infty} \frac{m_a n_a \kappa^2}{2\pi} \ln \left( \frac{|\mathbf{r}_j - \mathbf{r}_{j+i}|}{\xi} \right) \right]. \tag{3}$$

A finite core mass introduces a Newtonian kinetic-energy term into the standard Lagrangian of a many-vortex system, which is made of both a minimal-coupling-like and a potential-energy term ($\xi$ is a parameter having the dimensions of a length, typically of the order the core size, whose detailed value does not affect the equations of motion) [64, 66]. The relevant Euler-Lagrange equation for the $j^{\text{th}}$ vortex reads

$$\frac{M_c}{\kappa m_a n_a} \ddot{\mathbf{r}}_j = -\dot{\mathbf{r}}_j \times \hat{z} + \frac{\kappa}{2\pi} \sum_{i \in \mathbb{Z} \backslash \{0\}} \frac{\mathbf{r}_j - \mathbf{r}_{j+i}}{|\mathbf{r}_j - \mathbf{r}_{j+i}|^2}. \tag{4}$$

These equations admit as stationary solution the regular vortex configuration

$$\mathbf{r}_j(t) = (a j, 0), \quad \forall t. \tag{5}$$

In the most unstable mode, as depicted in Fig. 1, all the vortices are displaced from their equilibrium position according to

$$(a(j \pm i), 0) \quad \rightarrow \quad \left( a(j \pm i) + (-1)^i \epsilon_\parallel, (-1)^i \epsilon_\perp \right). \tag{6}$$

To develop the stability analysis of the fixed point (5), we linearize Eq. (4) with respect to the longitudinal (transverse) displacement $\epsilon_\parallel$ ($\epsilon_\perp$). The resulting system of two coupled second-

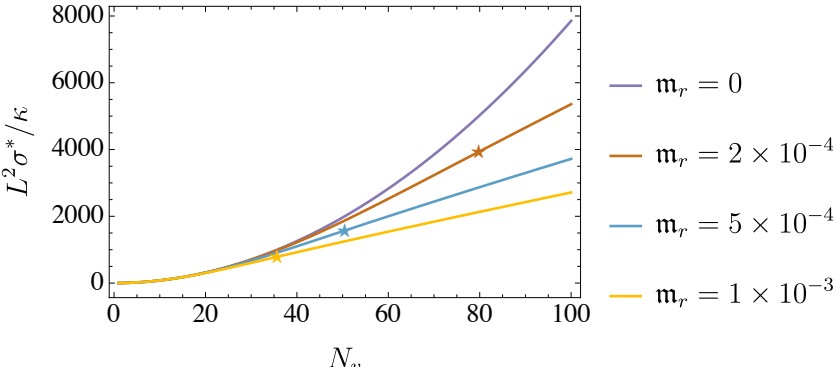

Figure 2: Dependence of the maximum instability growth rate $\sigma^*$, as given by Eq. (8), on the number of vortices $N_v$ along a row of length $L = aN_v$, for different values of the core mass, parametrized by $\mathfrak{m}_r = M_c/(m_a n_a L^2)$. Stars denote the critical values (10) at which the scaling $\sigma^*(N_v)$ switches from quadratic to linear.

order differential equations,

$$
\frac{M_c}{\kappa m_a n_a}\ddot{\epsilon}_\parallel = -\dot{\epsilon}_\perp - \frac{\pi\kappa}{4a^2}\epsilon_\parallel,
$$
$$
\frac{M_c}{\kappa m_a n_a}\ddot{\epsilon}_\perp = +\dot{\epsilon}_\parallel + \frac{\pi\kappa}{4a^2}\epsilon_\perp,
$$

(7)

admits solutions of the type $\epsilon_\parallel, \epsilon_\perp \sim e^{\lambda t}$ [73]. Among the possible values of $\lambda$, we focus on the one having the largest real part,

$$
\sigma^* = \frac{\kappa m_a n_a}{M_c\sqrt{2}}\sqrt{-1 + \sqrt{1 + \left(\frac{M_c\pi}{2a^2 m_a n_a}\right)^2}},
$$

(8)

as it constitutes the maximum instability growth rate. The latter quantity is shown in Fig. 2 as a function of the number of vortices $N_v$ contained in a length $L$, i.e. $N_v = L/a$. In the limit of massless cores ($M_c \to 0$) we recover the result of Eq. (2), i.e., $\sigma^* \propto a^{-2} \propto (N_v/L)^2 = (\Delta v/\kappa)^2$, where we recall that $\Delta v$ is the velocity difference across the vortex row. Such a quadratic dependence on $\Delta v$ is also found in the analysis of the classical Kelvin-Helmholtz instability, and keeps holding for small number $N_v$ of massive vortices. Remarkably, however, for large $N_v$ (i.e., small $a$) massive cores feature a maximum instability growth rate which is linear in $N_v$:

$$
\sigma^* \sim \sqrt{\frac{\pi n_a m_a}{M_c}}\frac{\kappa}{2L}N_v,
$$

(9)

or equivalently $\sigma^* \propto \Delta v$.

After Taylor-expanding Eq. (8) to second order in $M_c$, one obtains the critical value at which the scaling relation $\sigma^*(N_v)$ crosses over from quadratic to linear,

$$
\tilde{N}_v = 2L\sqrt{\frac{m_a n_a}{\pi M_c}},
$$

(10)

which is denoted by stars in Fig. 2. $\tilde{N}_v$ diverges in the limit of small values of $M_c$, recovering the asymptotic quadratic scaling $\sigma^*(N_v)$ for massless vortices.

Somehow reminiscently, the dispersion $\omega(k)$ of low-momentum elementary excitations in a Bose gas switches from quadratic to linear when introducing interactions between the bosons.

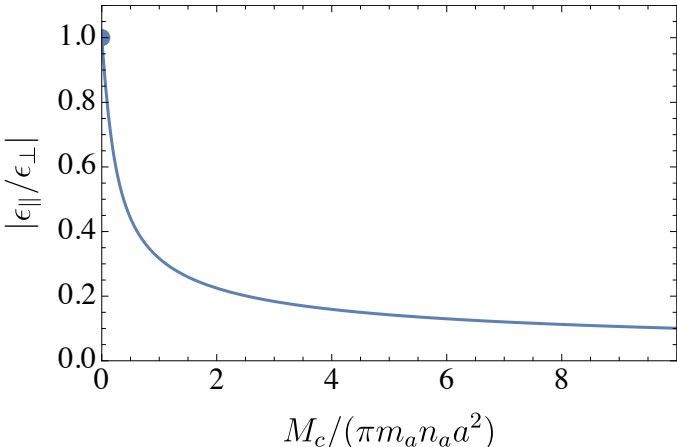

Figure 3: Ratio between the amplitudes of the longitudinal and transverse displacements for the most unstable eigenmode as a function of the core mass.

Both phenomena are non-perturbative, and may be seen as an occurrence of a *singular perturbation*. In the case of a Bose gas, interactions between bosons change the governing equation from linear (Schrödinger) to non-linear (Gross-Pitaevskii). In the present case, instead, the change of scaling of $\sigma^*$ may be traced back to the fact that the the dynamical equation governing the vortex motion is of first order in the case of massless vortices, but it becomes a second order equation for massive ones.

## 2.2 Longitudinal and transverse instability

Introducing the vector $\epsilon = \left(\epsilon_\parallel, \dot{\epsilon}_\parallel, \epsilon_\perp, \dot{\epsilon}_\perp\right)^T$, the two second-order equations of motion (7) can be recast into four first-order ordinary differential equations, written in matrix form as $\dot{\epsilon} = \mathbb{M}\epsilon$. The maximum instability growth rate $\sigma^*$ defined in Eq. (8) then corresponds to one of the eigenvalues of the $4 \times 4$ matrix $\mathbb{M}$ [73], with associated eigenvector

$$\mathbf{v}_{\sigma^*} = \left( \frac{\kappa m_a n_a}{M_c}\sigma^*, \frac{\kappa m_a n_a}{M_c}\sigma^{*2}, -\left(\sigma^{*2} + \frac{m_a n_a \pi \kappa^2}{4a^2 M_c}\right), -\sigma^*\left(\sigma^{*2} + \frac{m_a n_a \pi \kappa^2}{4a^2 M_c}\right)\right)^T. \quad (11)$$

We focus in particular on the ratio between the longitudinal and transverse components of the displacement vector

$$\frac{\epsilon_\parallel}{\epsilon_\perp} = \frac{v_{\sigma^*,1}}{v_{\sigma^*,3}}, \quad (12)$$

whose absolute value is shown in Fig. 3 as a function of the core mass $M_c$. In the massless case ($M_c = 0$), the most unstable mode has an equal longitudinal and transverse character, $|\epsilon_\parallel/\epsilon_\perp| = 1$. The ratio then monotonically decreases as the core mass is cranked up, meaning that the instability gets increasingly transverse.

## 2.3 Dissipation-induced suppression of the instability

The massive point vortex model (3) can be complemented to include dissipative processes that may hinder vortex motion. Surface tension and viscosity in classical fluids are two possible dissipation effects that reduce the growth rate, hence leading to a stabilization of the system. The dissipation channels that may open in superfluids have, instead, a different nature, as they can originate from a finite thermal component, density excitations [74] or Andreev vortex-bound states in the BCS regime [54,55].

According to the dissipative point-vortex model proposed by Schwarz, Kopnin, Sonin and others [75–80], quantized vortices scatter the elementary excitations of the superfluid and, in the presence of a non-zero relative velocity between the vortex and the gas of elementary excitations, an effective frictional force,

$$\mathbf{F}_j^N = \kappa m_a n_a \left[ d_\parallel (\mathbf{v}_n - \dot{\mathbf{r}}_j) + d_\perp \hat{z} \times (\mathbf{v}_n - \dot{\mathbf{r}}_j) \right], \tag{13}$$

acts on the vortex, $\mathbf{v}_n$ being the velocity of the normal fluid. In the following, we discuss the impact of this frictional force on the maximum instability growth rate $\sigma^*$ associated to regular vortex configurations.

Assuming a vanishing average velocity of the normal component ($\mathbf{v}_n = 0$), the equation of motion (4) for a massive vortex in an unbounded plane in presence of dissipation becomes

$$\frac{M_c}{\kappa m_a n_a} \ddot{\mathbf{r}}_j = -d_\parallel \dot{\mathbf{r}}_j + (1 - d_\perp) \hat{z} \times \dot{\mathbf{r}}_j + \frac{\kappa}{2\pi} \sum_{i \in \mathbb{Z} \setminus \{0\}} \frac{\mathbf{r}_j - \mathbf{r}_{j+i}}{|\mathbf{r}_j - \mathbf{r}_{j+i}|^2} . \tag{14}$$

While the longitudinal frictional term (with coefficient $d_\parallel$) is generally positive and therefore acts to slow down vortex motion, the transverse one (with coefficient $d_\perp$) is typically negative and therefore strengthens the Lorentz-like force $\propto \hat{z} \times \dot{\mathbf{r}}_j$. These equations are trivially satisfied by the stationary regular vortex row (5), because the latter is a static (or mechanical) equilibrium configuration, and hence it is not affected by the velocity-dependent dissipative force (13). For small oscillations around the equilibrium, linearizing Eq. (14) gives, in components:

$$
\begin{aligned}
\frac{M_c}{\kappa m_a n_a} \ddot{\epsilon}_\parallel &= -d_\parallel \dot{\epsilon}_\parallel - (1 - d_\perp) \dot{\epsilon}_\perp - \sigma_0^* \epsilon_\parallel , \\
\frac{M_c}{\kappa m_a n_a} \ddot{\epsilon}_\perp &= -d_\parallel \dot{\epsilon}_\perp + (1 - d_\perp) \dot{\epsilon}_\parallel + \sigma_0^* \epsilon_\perp ,
\end{aligned}
\tag{15}
$$

where $\sigma_0^*$ is the maximum instability growth rate for the massless and dissipationless case given in Eq. (2). The maximum instability growth rate $\sigma^*$ of the system corresponds to the largest real part,

$$\sigma^* = \max_{j \in [1,4]} \left\{ \Re \left( \lambda_j \right) \right\} , \tag{16}$$

among the four solutions of the characteristic equation associated to Eqs. (15),

$$\lambda^4 + 2\gamma d_\parallel \lambda^3 + \gamma^2 \left[ d_\parallel^2 + (1 - d_\perp)^2 \right] \lambda^2 - \left( \gamma \sigma_0^* \right)^2 = 0 , \tag{17}$$

where $\gamma = \kappa m_a n_a / M_c$.

To understand the effect of dissipation in the *massless* case, one can set $M_c = 0$ directly in Eqs. (15). By doing so, the second time derivatives drop out and one is left with a first-order problem for the displacements. In this case, the characteristic equation (17) simplifies and one can analytically determine the maximum instability growth rate

$$\sigma^* = \frac{\sigma_0^*}{\sqrt{d_\parallel^2 + (1 - d_\perp)^2}} . \tag{18}$$

The Taylor expansion $\sigma^* = \sigma_0^* \left( 1 + d_\perp - d_\parallel^2/2 + \dots \right)$ shows that, already in the massless case, non-zero mutual-friction coefficients $d_\parallel > 0$ and $d_\perp < 0$ cause a suppression of the system instability, with a first-order correction given by $d_\perp$, while $d_\parallel$ enters only at second order. The rate given by Eq. (18) is plotted in the left panel of Fig. 4.

A similar contour plot is shown in the right panel of Fig. 4 for the general case of $M_c > 0$, where the growth rate (16) comes out as a numerical solution of Eq. (17). The steeper slopes

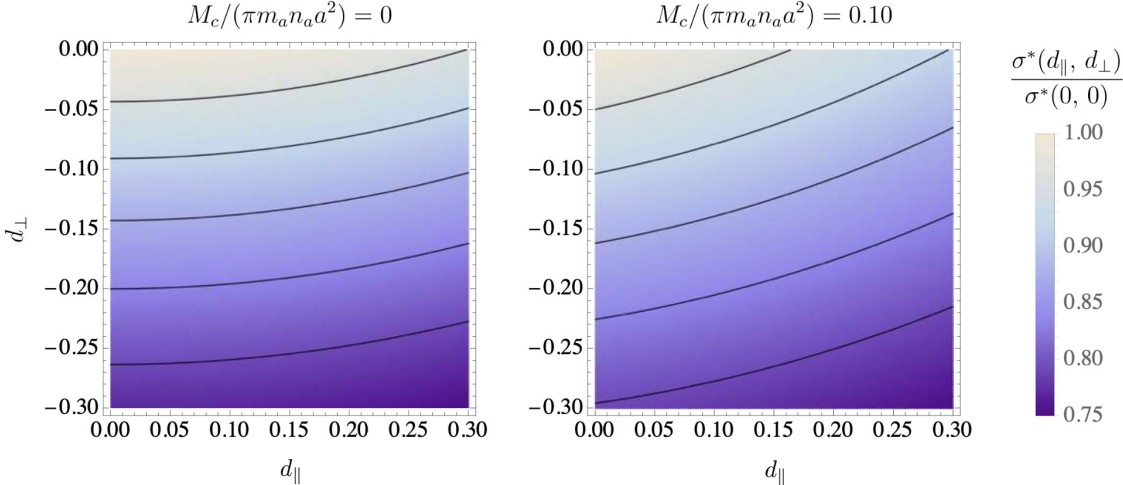

Figure 4: Contour plots of the maximum instability growth rate $\sigma^*$ of a vortex row as a function of the friction coefficients $d_\parallel$ and $d_\perp$. Left panel: plot of the massless result (18) normalized with respect to its dissipationless limit (2). Right panel: plot of the massive result (16) normalized with respect to its dissipationless limit (8).

of the contour lines between the two panels in Fig. 4 indicate that $\sigma^*$ becomes more sensitive to $d_\parallel$ in the presence of filled massive cores.

We conclude this section by observing that an alternative formulation of the mutual-friction force (13) acting on the $j$-th vortex is given by

$$\mathbf{F}_j^N = \kappa m_a n_a \left[ \alpha(\mathbf{v}_n - \mathbf{v}_{s,j}) - \alpha' \hat{z} \times (\mathbf{v}_n - \mathbf{v}_{s,j}) \right],\qquad(19)$$

where $\mathbf{v}_{s,j}$ is the superfluid velocity in the neighborhood of the vortex [77,79], and the longitudinal and transverse friction coefficients $\alpha$ and $\alpha'$ are both typically positive. In this framework, as was shown in Ref. [46], the maximum instability growth rate (18) reads

$$\sigma^* = \sigma_0^* \sqrt{\alpha^2 + (1 - \alpha')^2}.\qquad(20)$$

The Taylor expansion $\sigma^* = \sigma_0^* \left(1 - \alpha' + \alpha^2/2 + \dots\right)$ indicates an increase of the instability rate with longitudinal friction (controlled by $\alpha$), which naively looks in contrast with what discussed above. However, as shown in Ref. [79], in the absence of mass, the friction coefficients $\{\alpha, \alpha'\}$ are related to $\{d_\parallel, d_\perp\}$ by the relations $\alpha = d_\parallel / \left[ d_\parallel^2 + (1 - d_\perp)^2 \right]$, $\alpha' = 1 - (1 - d_\perp) / \left[ d_\parallel^2 + (1 - d_\perp)^2 \right]$, and using them, one may directly verify that Eq. (20) is completely equivalent to Eq. (18).

## 3  Massless vortex necklaces in annular superfluids

Vortex necklaces, also termed as vortex polygons, emerge at the interface between two counterflowing annular-like superfluids. The latter are particularly noteworthy for experimental protocols, as they lend themselves to implementing flows with periodic boundary conditions.

### 3.1  Point-vortex model

The dynamics of quantized vortices in a two-dimensional incompressible superfluid confined in an annular domain has been extensively studied in the recent Ref. [68] by means of a suitable

point-vortex model. While we refer the reader to that reference for an exhaustive derivation of such a model, in this section we review its main features.

The effective Lagrangian governing the dynamics of a system of $N_v$ point vortices of positive unit charge in a superfluid of uniform two-dimensional number density $n_a$ and atomic mass $m_a$ confined in an annulus of inner radius $R_1$ and outer radius $R_2$ reads:

$$\mathcal{L}_a = \sum_{j=1}^{N_v} \left\{ \pi \hbar n_a \left( R_2^2 - r_j^2 \right) \dot{\theta}_j - \Phi_j - \sum_{k=1}^{N_v}{}' V_{jk} \right\}, \tag{21}$$

where

$$\Phi_j = \Phi(r_j) \equiv \frac{\pi \hbar^2 n_a}{m_a} \left[ (1 - 2\mathfrak{n}_1) \ln\left( \frac{r_j}{R_2} \right) + \ln\left( \frac{2}{i} \frac{\vartheta_1\left( -i \ln(\frac{r_j}{R_2}), q \right)}{\vartheta_1'(0, q)} \right) \right], \tag{22}$$

is the one-vortex energy arising from the interaction of the vortex at $\boldsymbol{r}_j = (r_j, \theta_j)$ with its infinitely many images, while the two-vortex energy,

$$V_{jk} = V(\boldsymbol{r}_j, \boldsymbol{r}_k) \equiv \frac{\pi \hbar^2 n_a}{m_a} \operatorname{Re} \left\{ \ln \left[ \frac{\vartheta_1\left( \frac{1}{2}(\theta_j - \theta_k) - \frac{i}{2} \ln\left( \frac{r_j r_k}{R_2^2} \right), q \right)}{\vartheta_1\left( \frac{1}{2}(\theta_j - \theta_k) - \frac{i}{2} \ln\left( \frac{r_j}{r_k} \right), q \right)} \right] \right\}, \tag{23}$$

accounts for the interaction between vortices at position $\boldsymbol{r}_j$ and $\boldsymbol{r}_k$, including all their images (the primed sum means that the terms $k = j$ are omitted). The use of the Jacobi elliptic theta functions $\vartheta_1(z, q)$, which are integral functions of the complex variable $z$ and which depend also on the geometric ratio $q \equiv R_1/R_2$, allows for an exact treatment of the infinitely many image vortices ensuing from the presence of the two circular boundaries. Moreover, $\mathfrak{n}_1 \in \mathbb{Z}$ denotes the number of quanta of circulation around the inner boundary of the annulus. Figure 5 illustrates the particular case of a many-vortex system ($N_v = 6$) characterized by a regular arrangement of the vortices, lying on a circle of radius $r_0$ at equal distance one from the other.

The system has two conserved quantities:

- the total energy

$$\mathcal{H}_a = \sum_{j=1}^{N_v} \left( \Phi_j + \sum_{k=1}^{N_v}{}' V_{jk} \right), \tag{24}$$

which depends on the vortex positions $\{\mathbf{r}_j\}$, but not on their velocities $\{\dot{\mathbf{r}}_j\}$. This is a consequence of the first-order dynamics of 2D quantum vortices, whose $x$ and $y$ components play the role of canonically conjugate variables;

- the $z$-component of the angular momentum

$$L_a^z = \pi \hbar n_a \left( R_2^2 - R_1^2 \right) \mathfrak{n}_1 + \pi \hbar n_a \sum_{j=1}^{N_v} \left( R_2^2 - r_j^2 \right), \tag{25}$$

which includes the constant contribution from the possible non-zero circulation $\mathfrak{n}_1$ around the inner boundary and the sum $\sum_{j=1}^{N_v} \partial \mathcal{L}_a / \partial \dot{\theta}_j$ of the canonical angular momenta associated to the $N_v$ vortices. This integral of motion originates from the rotational invariance of the system.

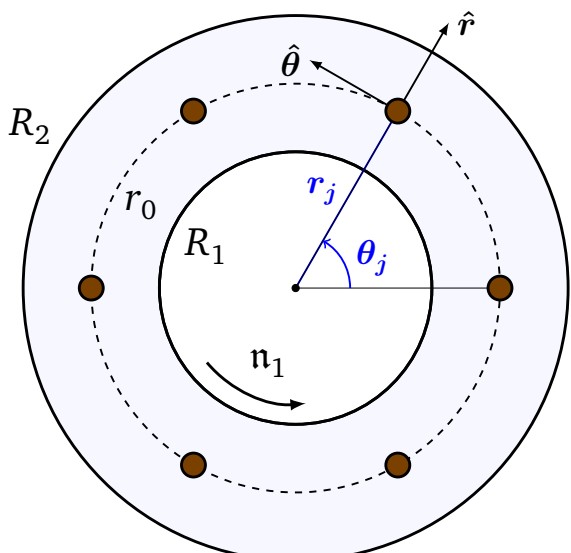

Figure 5: Schematic representation of the physical system for $N_v = 6$ vortices forming a regular necklace with radius $r_0$. The superfluid (light blue region) is confined in a two-dimensional annular domain having radii $R_1 < R_2$ and quantized flow circulation $\mathfrak{n}_1$ around the inner boundary. It hosts vortices with unit positive charge at positions $\boldsymbol{r}_j = (r_j, \theta_j)$.

## 3.2 Necklace solutions

A notable class of solutions of the Euler-Lagrange equations associated to Lagrangian (21) corresponds to regular vortex necklaces. As pictorially represented in Fig. 5, these are vortex structures of the type $r_j(t) = r_0$ and $\theta_j(t) = 2\pi j/N_v + \Omega^0_{N_v} t$, with $j = 1, 2, \ldots, N_v$, where

$$\Omega^0_{N_v}(r_0) = \frac{\hbar}{m_a r_0^2} \left[ \mathfrak{n}_1 - \frac{1}{2} + \frac{i}{2} \sum_{j=1}^{N_v} \frac{\vartheta'_1 \left( \frac{\pi}{N_v}(1-j) - i \ln\left( \frac{r_0}{R_2} \right), q \right)}{\vartheta_1 \left( \frac{\pi}{N_v}(1-j) - i \ln\left( \frac{r_0}{R_2} \right), q \right)} \right], \tag{26}$$

is the uniform-precession angular velocity of the whole system and $r_0$ is the radius of the $N_v$-vortex regular polygon. In Fig. 6, we plot Eq. (26) as a function of $r_0$ for different values of $N_v$.

Equation (26) generalizes and unifies various well-known results:

- When $r_0 = \sqrt{R_1 R_2}$, it reduces to

$$\Omega^0_{N_v}\left( r_0 = \sqrt{R_1 R_2} \right) = \frac{\hbar}{m_a r_0^2} \left[ \mathfrak{n}_1 + \frac{N_v - 1}{2} \right], \tag{27}$$

a formula that generalizes Eq. (B6) of Ref. [81], valid for a single vortex in an annulus, to the case of a $N_v$-vortex necklace. In the special case of $\mathfrak{n}_1 = -N_v/2$, the aforementioned expression reads

$$\Omega^0_{N_v}\left( r_0 = \sqrt{R_1 R_2} \right) = -\frac{1}{2} \frac{\hbar}{m_a r_0^2}, \tag{28}$$

and is therefore independent of the number of vortices.

- In the limit of an infinitely small internal boundary, it reduces to

$$\lim_{R_1 \to 0} \Omega^0_{N_v} = \frac{\hbar}{m_a r_0^2} \left[ \mathfrak{n}_1 + \frac{N_v - 1}{2} + N_v \frac{\left( \frac{r_0}{R_2} \right)^{2N_v}}{1 - \left( \frac{r_0}{R_2} \right)^{2N_v}} \right], \tag{29}$$

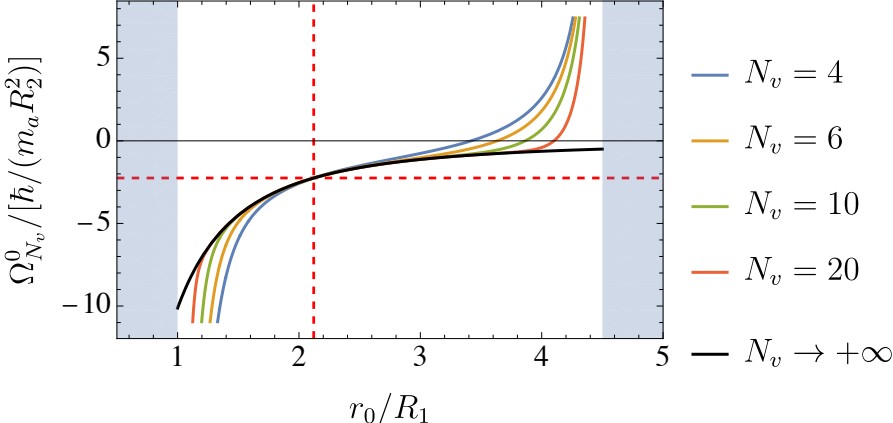

Figure 6: Precession frequency (26) as a function of $r_0$ for different values of $N_v$ and $\mathfrak{n}_1 = -N_v/2$, with $R_2 = 4.5R_1$. The vertical red dashed line corresponds to the geometric mean $\sqrt{R_1 R_2}$, while the horizontal red dashed line corresponds to Eq. (28). The blue-shaded rectangles correspond to regions which lie outside the annular domain.

and represents the precession frequency of an $N_v$-vortex necklace in a disk of radius $R_2$ in the presence of an extra vortex of charge $\mathfrak{n}_1$ at the disk's center. For $\mathfrak{n}_1 = 0$ this formula corresponds to the massless limit of Eq. (5) of Ref. [69].

- Taking the additional limit $R_2 \to +\infty$, the latter expression reduces to

$$\lim_{R_2 \to +\infty} \lim_{R_1 \to 0} \Omega_{N_v}^0 = \frac{\hbar}{m_a r_0^2} \left[ \mathfrak{n}_1 + \frac{N_v - 1}{2} \right], \tag{30}$$

which corresponds to Eq. (8) of Ref. [57], valid for a vortex necklace in the infinite plane.

Interestingly, Eq. (30) corresponds to Eq. (27), meaning that, when $r_0 = \sqrt{R_1 R_2}$, the precession frequency of the necklace in the annulus corresponds to that of a necklace in the unbounded plane. In fact, this special value of $r_0$ is such that the image charges generated by the two circular boundaries yield a net vanishing effect on the real vortices.

## 3.3 Instability of a massless vortex necklace

To perform the linear-stability analysis of vortex-necklace configurations, it is convenient to start by rewriting the Lagrangian (21) in a reference frame rotating at angular frequency $\Omega$. The coordinate transformation reads $r_j' = r_j$ and $\theta_j' = \theta_j - \Omega t$, where the primed variables are the ones in the rotating reference frame. The transformed Lagrangian,

$$\mathcal{L}_a' = \sum_{j=1}^{N_v} \left\{ \pi \hbar n_a \left( R_2^2 - r_j^2 \right) \left( \dot{\theta}_j' + \Omega \right) - \Phi_j - \sum_{k=1}^{N_v}{}' V_{jk} \right\}, \tag{31}$$

is such that both $\Phi_j$ and $V_{jk}$ are unaltered by the transformation. Comparing Eqs. (31) and (21), it is clear that the kinetic term $\mathcal{T}_a' = \pi \hbar n_a \sum_{j=1}^{N_v} (R_2^2 - r_j^2) \dot{\theta}_j'$ is formally unaltered, while the potential term is modified as follows:

$$\mathcal{H}_a \quad \to \quad \mathcal{H}_a' = \sum_{j=1}^{N_v} \left( \Phi_j + \sum_{k=1}^{N_v}{}' V_{jk} \right) - \Omega \sum_{j=1}^{N_v} \pi \hbar n_a (R_2^2 - r_j^2). \tag{32}$$

Provided that $\Omega$ equals $\Omega_{N_v}$ [as given by Eq. (26)], the $N_v$-vortex necklace constitutes a stationary configuration in the rotating reference frame.

We develop the linear-stability analysis according to the scheme described in Ref. [82], i.e. by diagonalizing the $2N_v \times 2N_v$ matrix

$$\mathbb{J} = t^{-1}\,\mathbb{S}\,\mathbb{H}\,, \tag{33}$$

where

$$t = \left(\frac{\partial^2 \mathcal{T}_a'}{\partial r_j' \, \partial \dot{\theta}_j'}\right)_{\text{eq}}, \tag{34}$$

is actually independent of $j$ and

$$(\mathbb{H})_{i,j} = \left(\frac{\partial^2 \mathcal{H}_a'}{\partial D_i \, \partial D_j}\right)_{\text{eq}}, \tag{35}$$

is the Hessian matrix associated to Hamiltonian (32) and evaluated at the regular-necklace configuration (hence the subscript "eq"). The vector $\mathbf{D} = \left(r_1', ..., r_{N_v}', \theta_1', ..., \theta_{N_v}'\right)^T$ constitutes the full array of dynamical variables in the rotating reference frame (recall that $r_j \equiv r_j'$). Eventually, the antisymmetric matrix,

$$\mathbb{S} = \begin{pmatrix} 0_{N_v} & I_{N_v} \\ -I_{N_v} & 0_{N_v} \end{pmatrix}, \tag{36}$$

encodes the Hamiltonian structure of the system, $0_{N_v}$ and $I_{N_v}$ being, respectively, the zero- and the identity matrix of order $N_v$.

The $2N_v$ eigenvalues of $\mathbb{J}$ are complex numbers, which we write as

$$\lambda_j = \sigma_j + i\,\omega_j\,. \tag{37}$$

As is well-known from the theory of Hamiltonian matrices, recall that also $-\lambda_j$, $\lambda_j^*$, and $-\lambda_j^*$ are eigenvalues of $\mathbb{J}$. Their imaginary parts are the frequencies of stable small oscillations around the necklace solution, while their real parts describe their instability. In the following, we will focus on the set of $\sigma_j$'s, which are often termed *instability growth rates*.

Figure 7 shows the instability growth rate $\sigma$ on the wavenumber $q_j = 2\pi j/(N_v d_v)$, for different values of $N_v$ (only one half of the full dispersion relation is shown, being it symmetric with respect to the most unstable wavenumber).

In all cases, the most unstable mode is the one associated to $q^* = q_{j=N_v/2} = \pi/d_v$, where $d_v \approx 2\pi r_0/N_v$ is the intervortex distance, while the mode $q = 0$ is such that $\sigma(q = 0) = 0$. The latter property is associated to the conservation of the total angular momentum, which follows from system rotational invariance. For $N_v \to +\infty$, the points collapse on Eq. (1), which was derived for straight vortex rows in Ref. [43]. The structure of the normal modes for $N_v = 6$ is illustrated in Fig. 8.

## 3.4 Boundary-induced stabilization of the necklace

The presence of the annulus boundaries contributes to stabilize a vortex necklace. Indeed, in Fig. 9 we show the maximum instability growth rate $\sigma^* := \max_{j \in [1, 2N_v]}\{\sigma_j\}$ [where the $\sigma_j$'s are given by Eq. (37)] as a function of the annulus width $\Delta R = R_2 - R_1$.

One can appreciate that, for any $N_v$, the necklace becomes stable on narrow enough annuli. The reason can be ascribed to the competition between two different length scales: the typical intervortex distance $d_v \approx 2\pi r_0/N_v$ and the distance between the vortices and the boundaries,

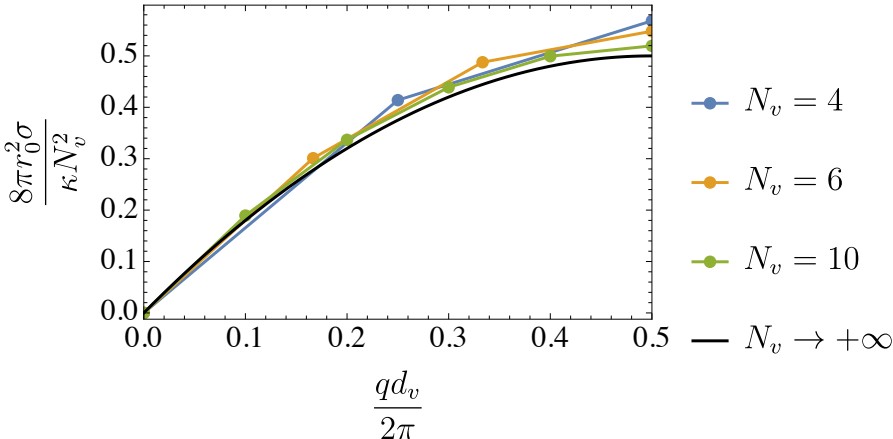

Figure 7: Dependence of the instability growth rates $\sigma$ [corresponding to the real parts of eigenvalues (37)] on the eigenmode wavenumber $q$ for different values of $N_v$. Results obtained for necklaces of radius $r_0 = \sqrt{R_1 R_2}$ and $\mathfrak{n}_1 = -N_v/2$. The black solid line corresponds to Eq. (1).

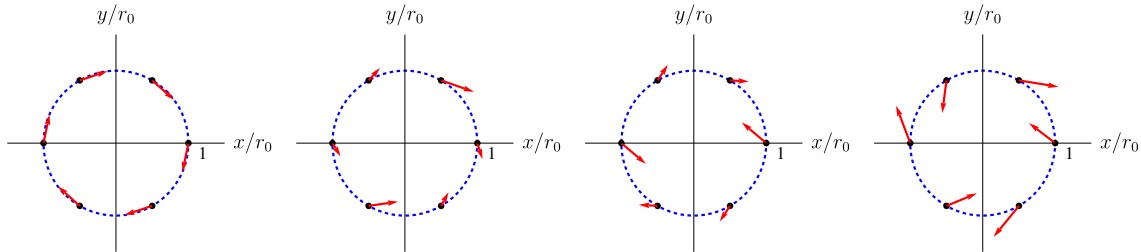

Figure 8: Structure of the 4 independent eigenmodes associated to a 6-vortex necklace. Panels correspond to the allowed wavenumbers $q = 0, 1, 2, 3$ (from left to right). Red arrows represent the relevant displacement vectors. The first panel shows the stable mode, while the last one is the most unstable (i.e., the one with $q = q^*$).

which is of the order $\Delta R$. In the limit of a fat annulus, $\Delta R \gg d_v$, the dynamics of each vortex is mainly determined by the remaining $N_v - 1$ physical vortices, while, in the opposite limit of a thin annulus $\Delta R \ll d_v$ such dynamics is mainly determined by the image vortices. The critical condition corresponding to the cross-over between the two aforementioned regimes is

$$\Delta R_c = d_v = \frac{2\pi r_0}{N_v} \, . \tag{38}$$

Moreover, if the annulus width $\Delta R$ tends to infinity, the maximum instability growth rate tends to

$$\lim_{\Delta R \to +\infty} \sigma^* = \frac{\hbar}{8 m_a r_0^2} N_v \sqrt{N_v^2 - 8 \left( N_v - 1 + 2\mathfrak{n}_1 \right)} \, . \tag{39}$$

Setting $\mathfrak{n}_1 = 0$, this formula yields the known result for a regular vortex polygon on an unbounded plane [Eq. (5.14b) of Ref. [43]]. Notice the scaling $\sigma^* \sim N_v^2$ for large values of $N_v$.

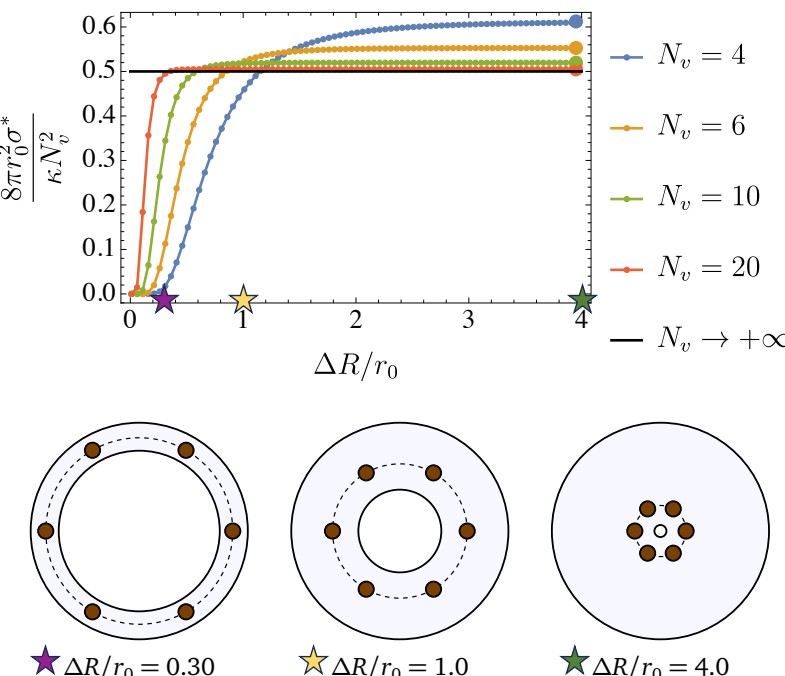

Figure 9: Dependence of the maximum instability growth rate $\sigma^*$ on the annulus width $\Delta R = R_2 - R_1$ for necklaces with $N_v$ vortices, at fixed $r_0 = \sqrt{R_1 R_2}$ and $\mathfrak{n}_1 = -N_v/2$. Vortex necklaces are more stable in the presence of narrower annuli. The thick dots represent the limit of large-width annuli, Eq. (39), while the black line is the result for a straight chain, Eq. (2). The sketches below the plot show three geometric configurations for a 6-vortex necklace.

## 4 Massive vortex necklaces in annular superfluids

In this section we show that also a non-zero core mass tends to stabilize a vortex necklace on an annular domain, thus unifying and generalizing the analyses developed in Secs. 2 and 3.

### 4.1 Hamiltonian description

When vortex cores have a mass $M_c$, the ensuing inertial contribution to the dynamics of each vortex can be easily introduced in the Lagrangian model [64, 66, 68], so that Eq. (21) modifies as follows

$$\mathcal{L}\left(\{r_j, \theta_j\}, \{\dot{r}_j, \dot{\theta}_j\}\right) = \sum_{j=1}^{N_v} \left\{ \frac{M_c}{2}\left(\dot{r}_j^2 + r_j^2\dot{\theta}_j^2\right) + \pi\hbar n_a\left(R_2^2 - r_j^2\right)\dot{\theta}_j - \Phi_j - \sum_{k=1}^{N_v}{}' V_{jk} \right\}. \quad (40)$$

To carry out the linear-stability analysis, it is convenient to resort to an equivalent Hamiltonian description, where the total energy,

$$\mathcal{H} = \sum_{j=1}^{N_v} \left\{ \frac{p_{r_j}^2}{2M_c} + \frac{\left[p_{\theta_j} - \pi\hbar n_a\left(R_2^2 - r_j^2\right)\right]^2}{2M_c r_j^2} + \Phi_j + \sum_{k=1}^{N_v}{}' V_{jk} \right\}, \quad (41)$$

can be obtained upon a standard Legendre transform of Lagrangian (40) and constitutes the massive version of Hamiltonian (24). Notice that, if compared to the latter, Hamiltonian (41)

depends on twice as many dynamical variables, since the $2N_v$ independent canonical momenta

$$p_{r_j} = \frac{\partial \mathcal{L}}{\partial \dot{r}_j} = M_c \dot{r}_j, \tag{42}$$

$$p_{\theta_j} = \frac{\partial \mathcal{L}}{\partial \dot{\theta}_j} = M_c r_j^2 \dot{\theta}_j + \pi \hbar n_a \left( R_2^2 - r_j^2 \right), \tag{43}$$

are unlocked by the introduction of a non-zero $M_c$ [66].

The notable class of solutions described in Sec. 3.2 and corresponding to regular vortex necklaces is, in general, modified, as the core inertial mass results in an additional centrifugal force acting on each vortex. After defining the total mass of the superfluid component $M_a = \pi \left( R_2^2 - R_1^2 \right) n_a m_a$, and introducing the mass ratio $\mathfrak{m}_n = M_c / M_a$ (the subscript $n$ stands for "necklace"), one can compute the precession frequency,

$$\Omega_{N_v}(r_0) = \frac{\hbar}{m_a \left( R_2^2 - R_1^2 \right)} \frac{1}{\mathfrak{m}_n} \left( 1 - \sqrt{1 - 2\mathfrak{m}_n \frac{m_a \left( R_2^2 - R_1^2 \right)}{\hbar} \Omega_{N_v}^0(r_0)} \right), \tag{44}$$

of the massive $N_v$-vortex necklace. This expression involves the precession frequency $\Omega_{N_v}^0$ of the massless vortex necklace, Eq. (26), and reduces to it in the limit $\mathfrak{m}_n \to 0$.

## 4.2 Core-mass-induced suppression of the instability

We show now that the presence of a finite mass filling the cores strongly affects the stability properties of vortex necklaces against perturbation.

To develop the linear-stability analysis, we preliminary rewrite Hamiltonian (41) in a (primed) reference frame rotating at frequency $\Omega_{N_v}$ with respect to the (unprimed) laboratory frame according to the standard transformation

$$\mathcal{H}\left(\{\mathbf{r}_j\}, \{\mathbf{p}_j\}\right) \quad \to \quad \mathcal{H}'\left(\{\mathbf{r}'_j\}, \{\mathbf{p}'_j\}\right) = \mathcal{H}\left(\{\mathbf{r}'_j\}, \{\mathbf{p}'_j\}\right) - \Omega_{N_v} \sum_{j=1}^{N_v} (\mathbf{r}'_j \times \mathbf{p}'_j) \cdot \hat{z} \tag{45}$$

(recall that $r_j \equiv r'_j$ by definition). Then, one computes the matrix $\mathbb{J} = \mathbb{S}\mathbb{H}$, where $\mathbb{H}$ is the Hessian matrix associated to Hamiltonian (45) and evaluates it at the rotating regular-necklace configuration, which constitutes a fixed-point (hence the subscript "eq") when observed from the rotating reference frame. Notice that, as opposed to the scheme illustrated in Sec. 3.3 and to Eq. (35), the vector

$$\mathbf{D} = \left( r'_1, ..., r'_{N_v}, \theta'_1, \ldots, \theta'_{N_v}, p'_{r_1}, \ldots, p'_{r_{N_v}}, p'_{\theta_1}, \ldots, p'_{\theta_{N_v}} \right)^T, \tag{46}$$

now includes twice as many dynamical variables, because the introduction of core mass doubles the dimension of the associated phase space. The antisymmetric matrix $\mathbb{S}$ is the $4N_v \times 4N_v$ version of Eq. (36). The $4N_v$ eigenvalues of $\mathbb{J}$ are of the type

$$\lambda_j = \sigma_j + i\,\omega_j, \tag{47}$$

and they determine the stability of the regular massive $N_v$-vortex necklace. As discussed in Sec. 3.4, we are primarily interested in

$$\sigma^* := \max_{j \in [1, 4N_v]} \left\{ \sigma_j \right\}, \tag{48}$$

as it is the rate that characterizes the breakdown of the necklace structure upon perturbation.

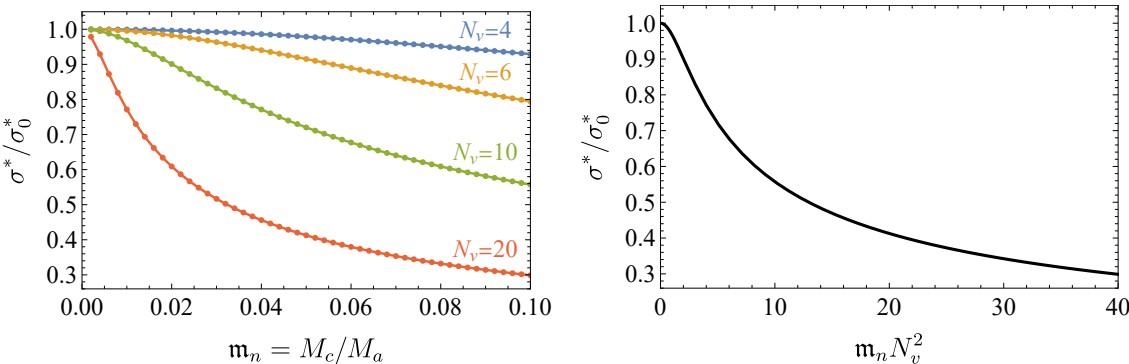

Figure 10: Maximum instability growth rate $\sigma^*$ (normalized to its value $\sigma_0^*$ at zero mass) for a massive vortex necklace of radius $r_0 = \sqrt{R_1 R_2}$ inside an annulus with radii $R_2 = 4.5 R_1$. Left: dependence of $\sigma^*$, as defined in Eq. (48), on the vortex core mass $M_c$, for different number of vortices $N_v$. Right: dependence of $\sigma^*$, as given by Eq. (49), on the parameter $\mathfrak{m} N_v^2$.

Quite generally, the core mass tends to *stabilize* the necklaces, i.e. the maximum instability growth rate $\sigma^*$ is smaller in the presence of core mass. In the left panel of Fig. 10 we illustrate this effect for necklaces featuring different values of $M_c$.

One can observe that the suppression of the instability is more effective for larger values of $N_v$. Additional insights into the physical mechanism responsible for this suppression can be obtained adapting Eq. (8), rigorously valid for an infinite vortex row, to the case of a $N_v$-vortex necklace of radius $r_0$. This is obtained via the substitution $a = 2\pi r_0 / N_v$, yielding the following maximum instability growth rate for a massive necklace:

$$\sigma^* = \sigma_0^* \frac{r_0^2}{R_2^2 - R_1^2} \frac{8\sqrt{2}}{\mathfrak{m}_n N_v^2} \sqrt{-1 + \sqrt{1 + \left( \frac{\mathfrak{m}_n N_v^2}{8} \frac{R_2^2 - R_1^2}{r_0^2} \right)^2}}. \tag{49}$$

This relation, illustrated in Fig. 11 as a function of the number of vortices (solid lines), well captures the results of the full numerical linear-stability analysis (dots) for a vortex necklace in an annular domain.

Interestingly, the quantity $\mathfrak{m}_n N_v^2$ emerges as a universal parameter in the above Eq. (49). In fact, the different curves in the left panel of Fig. 10 eventually collapse onto a single curve $\sigma^*(\mathfrak{m}_n N_v^2)$ in the right panel.

As already shown in Sec. 2.1, the introduction of a non-zero core mass modifies the asymptotic behaviour of $\sigma^*(N_v)$: the *quadratic* scaling law,

$$\sigma^* = \frac{\kappa}{16\pi r_0^2} N_v^2, \tag{50}$$

characterizing the well-known massless scenario gives way to the *linear* dependence

$$\sigma^* \sim \sqrt{\frac{m_a n_a}{M_c \pi}} \frac{\kappa}{4 r_0} N_v. \tag{51}$$

Comparing Eqs. (50) and (51), one can easily determine the critical value

$$\tilde{N}_v = 4 \sqrt{\frac{\pi m_a n_a r_0^2}{M_c}}, \tag{52}$$

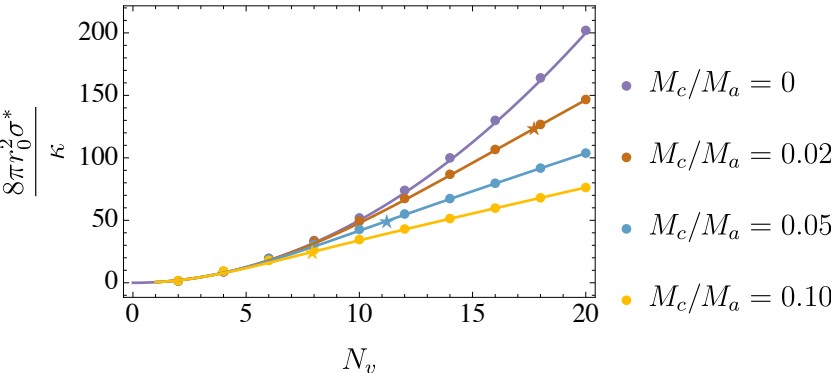

Figure 11: Dependence of the maximum instability growth rate $\sigma^*$ on the number of vortices $N_v$ for different values of the core mass $M_c$. Dots correspond to the results of the full linear-stability analysis of a vortex necklace in an annular domain [see Eq. (48)], while solid lines corresponds to the predictions of Eq. (8) adapted to the case of vortex necklaces. Stars denote the critical values (52) at which the scaling $\sigma^*(N_v)$ crosses over from quadratic to linear.

below (above) which the dependence of $\sigma^*$ on $N_v$ is quadratic (linear), and observe that, for $M_c \to 0^+$, it diverges, a circumstance which confirms the asymptotic quadratic dependence characterizing massless vortices (see Fig. 11). This change of scaling law can be equivalently formulated in terms of the velocity difference,

$$\Delta v = \frac{\kappa}{2\pi r_0} N_v \,, \tag{53}$$

at the interface between the two counter-propagating flows. One can verify, in fact, that the well-known quadratic scaling $\sigma^* \propto \Delta v^2$ associated to Eq. (50) and typical of the classical [5, 16] as well as of the superfluid Kelvin-Helmholtz instability [43, 46] is replaced by the linear scaling $\sigma^* \propto \Delta v$ associated to Eq. (51).

## 4.3 Azimuthal and radial instability

Both the radial ($\delta r_j$) and the azimuthal ($r_0 \delta \theta_j$) displacements of the $j^{\text{th}}$ vortex with respect to their respective equilibrium values diverge as $\sim e^{\sigma^* t}$ in the neighbourhood of the fixed-point configuration. Moreover, as previously discussed, the vortex-necklace instability can be further characterized through the ratio $\epsilon_\parallel / \epsilon_\perp := r_0 \delta \theta_j / \delta r_j$. Such a quantity is easily computed from the entries of the eigenvector associated to $\sigma^*$, along the same lines as Eqs. (11, 12). The (absolute value of) ratio $\epsilon_\parallel / \epsilon_\perp$ is shown in Fig. 12 as a function of the number of vortices in the necklace, for different values of the core mass. In the massless case (uppermost curve), the quantity saturates to 1 for a large number of vortices, meaning that the perturbation has equal radial and azimuthal components. The larger the core mass, the smaller the ratio, signaling that the radial character of the instability prevails. This scenario, showing that the finite core mass enhances the transverse nature of the instability, is consistent with the one discussed in relation to the linear vortex row (see Sec. 2.2 and, in particular, Fig. 3).

## 5 Comparison with experiments

Recently, the superfluid Kelvin-Helmholtz instability has been observed with unprecedented accuracy in an atomic superfluid of $^6$Li [46]. Upon imprinting two counter-rotating flows with

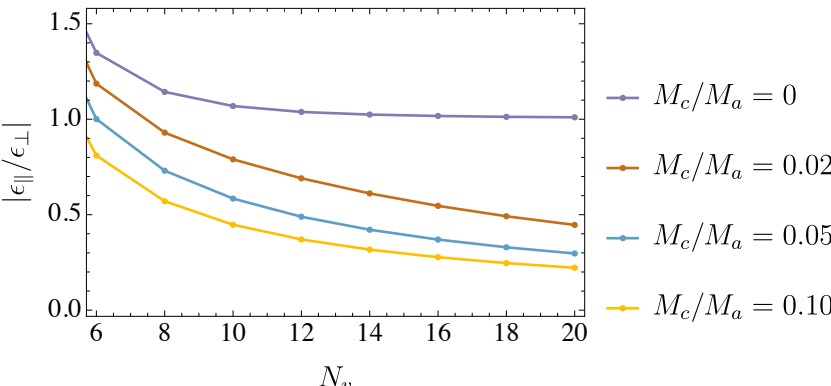

Figure 12: Ratio between the amplitude of the tangential displacement and that of the radial displacement for the most unstable eigenmode. Upon increasing the core mass $M_c$, the instability, in the limit of large $N_v$, becomes increasingly radial, i.e. transverse to the necklace.

tunable relative velocity, the LENS group characterized the development of an ordered vortex necklace and its subsequent breakdown due to the onset of instability. In this section, we compare their experimental measurements, performed across different regimes, ranging from weakly-interacting bosonic (BEC side) to strongly-correlated fermionic pair condensate (BCS side), with the predictions of our dissipative and massive point-vortex model.

Unfortunately, it is not possible to rigorously perform the linear-stability analysis for a (either massless or massive) vortex necklace subject to frictional forces. The reason is that, in the presence of dissipation, regular vortex necklaces (see Sec. 3.2 and Sec. 4.1) no longer constitute stationary solutions of the associated dynamical systems. This circumstance prevents the application of the standard linear-stability-analysis machinery described in Sec. 3.3 and Sec. 4.2 and would require a systematic analysis of the early-time dynamics generated by the full set of equations of motion with suitable stochastic initial conditions. However, assuming that the breakdown of the necklace is only due to the Kelvin-Helmholtz mechanism, one can estimate the corresponding instability growth rate in the presence of dissipation adapting Eqs. (16)-(18), obtained for a vortex row, to the case of a vortex necklace. This can be done upon the substitution $a = 2\pi r_0/N_v$ and, on the basis of what was discussed in Sec. 4.2 (and illustrated in Fig. 11), the resulting equations are expected to well capture the properties of the actual vortex necklace.

The outcomes of this analysis are shown in Fig. 13 as a function of the (squared) velocity difference (53) at the interface between the two counter-rotating flows. According to the results of Ref. [54] (see, in particular, panel 1h), we took $\alpha = 0.01$ as a realistic estimate of the longitudinal mutual-friction coefficient. Moreover, one can introduce the parameter

$$f = \frac{M_c}{\pi \xi^2 m_a n_a}, \tag{54}$$

representing the filling fraction of a vortex core of radius $\xi \sim 0.75$ $\mu$m, that, for this specific experimental platform, has two natural bounds: 0 (completely empty core), and 1 (density of quasi-particle bound states equal to the superfluid density).

As visible in the figure, the transverse mutual-friction coefficient $\alpha'$ can indeed cause a significant suppression of the instability. While experimental data on this coefficient are currently lacking, our findings suggest that the measured values of $\sigma^*$, significantly smaller than the expected one [see Eq. (2)], would be reproduced if $\alpha' \approx 0.75$.

Moreover, for the current system, the presence of core mass up to $f = 1$ appears to have a negligible impact on the relation $\sigma^*(\Delta v^2)$. We have verified that the core-mass-induced

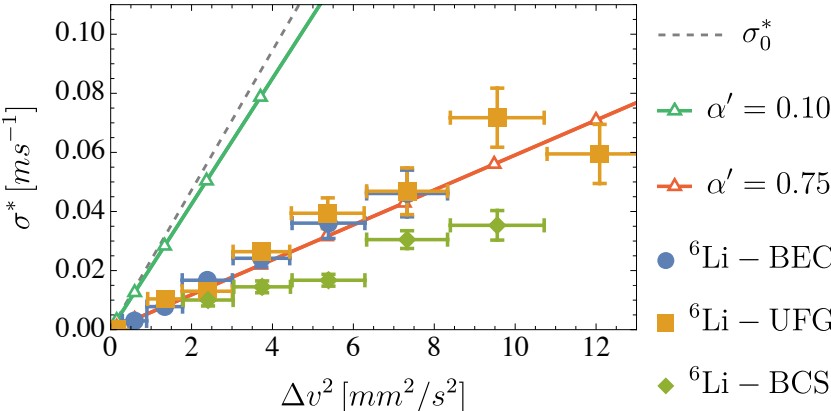

Figure 13: Dependence of the maximum instability growth rate $\sigma^*$ on the velocity difference at the interface between two counter rotating flows. The gray dashed line corresponds to Eq. (2), written in terms of $\Delta v$. Filled blue circles, orange squares, and green diamonds represent the experimental measurements reported by the LENS group for an atomic superfluid of $^6$Li atoms (data extracted from Fig. 3f of Ref. [46]). The green and the red solid lines represent the predictions of our generalized point-vortex model, for $\alpha' = 0.10$ and $\alpha' = 0.75$, respectively. In both cases we assumed $\alpha = 0.01$, a value which is compatible with the results reported in Fig. 1h of Ref. [54]. There is no visible difference between the predictions for $f = 0$ (solid lines) [see Eq. (20)] and the predictions for $f = 1$ (open triangles) [see Eq. (16)].

suppression of such scaling indeed occurs, but only at much larger ($f \sim 50$), and therefore unrealistic, values of the filling fraction (54). The filling fraction $f = 1$ corresponds to the tiny mass ratio $M_c/M_a \simeq 10^{-4}$, thus explaining the absence of a visible effect on $\sigma^*$ in Fig. 13. We would mention that the filling of vortex cores can substantially increase in the deep BCS regime, where a larger number of quasi-particle bound states could provide the vortices with a non-negligible core mass (up to $M_c/M_a \simeq 10^{-2}$). Also, in a very recent study [41] by An et al., it was found that in the onset of the superfluid Kelvin-Helmholtz instability at the interface of two distinct superfluids, vortices within each superfluid are populated by particles from the other superfluid. Consequently, all vortices in that system are endowed with a sizable inertial mass (and hence to a filling fraction $f$ significantly larger than zero).

In summary, we have shown that the introduction of core mass and dissipative effects into the point-vortex model is, in general, responsible for a stabilization of vortex necklaces. From a quantitative perspective the main role seems to be played by the transverse mutual-friction coefficient $\alpha'$.

## 6  Conclusions

In this work, we analyzed the stability of vortex rows and vortex necklaces which typically appear at the interface between two superflows having a non-zero relative velocity (hence the analogy with the well-known classical Kelvin-Helmholtz instability). Previous theoretical works [43, 56, 57] have quantified the instability growth rate for certain noteworthy classes of point-vortex structures in ideal fluids. However, none of these works considered the additional effects, such as finite vortex core mass and dissipation, which are often present in real superfluid systems and may influence the breakdown of such structures.

In Sec. 2, we studied the effect of a finite vortex core mass on the linear-stability analysis of vortex rows. This is motivated by the fact that, in many real experimental platforms, quantum vortices are often filled by massive particles, either deliberately or accidentally. We showed that, in general, vortex rows exhibit increased resilience to the onset of dynamical instabilities when core mass is considered. Interestingly, the introduction of a finite core mass affects the dependence of the maximum instability growth rate ($\sigma^*$) on the number of vortices per unit length ($N_v/L$), as its scaling law changes from a quadratic to linear behaviour. Moreover, we pointed out that, while in the massless case the instability develops along the longitudinal and the transverse direction in equal measure, in the presence of massive cores, vortices depart from their regular configuration mainly in a transverse fashion. We have also introduced an additional and physically-relevant process into our linear-stability analysis, dissipation, and highlighted its stabilizing effect on vortex rows. Surprisingly, our analysis revealed that the suppression of the instability is mainly due to $d_\perp$, while it only moderately depends on $d_\parallel$.

In Sec. 3, building upon the model that we introduced to investigate the dynamics of many-vortex systems in annular domains [68], we analyzed the (in)stability properties of vortex necklaces, showing that they can be even stabilized if confined in narrow annular domains. This detailed analysis is motivated by the fact that narrow annuli and, more in general, narrow channels along which superfluids flow, constitute the typical setups used to investigate the development of the superfluid Kelvin-Helmholtz instability [42, 44, 46].

In Sec. 4, we delved into the influence of a finite vortex core mass on the stability properties of vortex necklaces. Our investigation revealed a significant role played by the inertia of vortex cores in suppressing the necklace instability. This phenomenon came with other notable changes, including a shift in the asymptotic scaling of $\sigma^*(N_v)$ from quadratic to linear, and a transformation of the most unstable eigenmode towards increased transverse behavior.

In conclusion, we have elucidated the role played by confinement, vortex mass and mutual friction on the dynamical instability of many-vortex systems. Further studies are needed to understand the role of mutual friction in the breakdown of rotating vortex necklaces. This should require the explicit solution of the full set of equations of motion with suitable stochastic initial conditions, thus providing a more reliable estimate of $\sigma^*$ for the case of a rotating necklace in a dissipative superfluid. Moreover, it could be interesting to analyze other effects which may affect the Kelvin-Helmholtz instability, such as non-zero temperatures, unequal vortex-core mass [83], or the coupling of vortices to sound.

## Acknowledgments

We acknowledge insightful discussions with C.F. Barenghi, A.L. Fetter, D. Hernandez-Rajkov, F. Marino, M. Modugno, G. Roati, and F. Scazza.

**Funding information**  A. R. received funding from the European Union's Horizon research and innovation programme under the Marie Skłodowska-Curie grant agreement *Vortexons* (no. 101062887). P. M. and A. R. acknowledge support by the Spanish Ministerio de Ciencia e Innovación (MCIN/AEI/10.13039/501100011033, grant PID2020-113565GB-C21), and by the Generalitat de Catalunya (grant 2021 SGR 01411). P. M. further acknowledges support by the *ICREA Academia* program. M. Cal., A. R. and P. M. would like to thank the Institut Henri Poincaré (UAR 839 CNRS-Sorbonne Université) and the LabEx CARMIN (ANR-10-LABX-59-01) for their support.

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
