# Peer review of "Suppression of the superfluid Kelvin-Helmholtz instability due to massive vortex cores, friction and confinement"

_SciPost Physics, doi:SciPost Phys. 17, 076 (2024)_

## Round 1 · Referee Report · Anonymous (Referee 1) · 2024-5-25

Strengths

The manuscript report a theoretical investigation of the instability of arrays of quantized vortices appearing at the interface between two superfluid layers in relative motion. Recent theoretical [42, 44] and experimental [46] studies have explored these dynamics, uncovering intriguing connections to the well-known Kelvin-Helmholtz instability of a shear layer in irrotational inviscid fluids.

1) The authors address the problem by introducing a generalized point-vortex model to include massive vortex cores and dissipative processes and they show that these effects can lead to the partial or complete suppression of the instability.

2) The manuscript is well-written, scientifically sound and, in my opinion, holds significant interest for the community as it addresses the role played by effects which have yet to be fully understood in real superfluids.

Weaknesses

1) Several interesting aspects raised in the manuscript require further discussion.

2) A couple of technical points should be clarified (See report)

Report

The manuscript report a theoretical investigation of the instability of arrays of quantized vortices appearing at the interface between two
superfluid layers in relative motion. Recent theoretical [42, 44] and experimental [46] studies have explored these dynamics, uncovering intriguing connections to the well-known Kelvin-Helmholtz instability of a shear layer in irrotational inviscid fluids. From a purely theoretical perspective, the problem of the stability of arrays of identical point vortices can be traced back to the works of Rosenhead and Havelock in 1931. More recently, Aref [43] expanded on these ideas, viewing the array as a discrete version of a vortex sheet, thereby establishing a direct conceptual connection with Kelvin-Helmholtz scenario.

Here, the authors address the problem of the stability of the shear layer at the interface between counter-propagating superfluids,
demonstrating that massive vortex cores and dissipative processes, together with the proximity with the boundaries of the sample,
are responsible for a partial or complete quenching of the instability. The analysis is based on a generalized point-vortex model, including
the effects of a finite vortex-core mass and dissipation.

The manuscript is well-written, scientifically sound and, in my opinion, holds significant interest for the community as it addresses
the role played by effects which have yet to be fully understood in real superfluids and may influence the stability properties of
arrays of quantized vortices. For these reasons, I think that the work potentially deserves publication. However, some issues need to be addressed and clarified. These issues are reported below in the order they arise in the text, not by their importance.

Requested changes

1) In the introduction the authors state that "The kinematic viscosity $\mu$ introduces an ultra-violet cutoff that halts the runaway behaviour for small wavelengths (i.e. large $q$).

This is only partially correct. In classical hydrodynamics, an ultraviolet cutoff arises whenever a singular distribution of vorticity, i.e., a zero-thickness shear layer, is no longer an accurate description of the system (see, e.g. Ref. [5]). A more realistic approach is to consider a finite-width shear layer that continuously connects the two uniform flows. While it is true that viscous effects can make the zero-thickness shear-layer approximation less accurate, in general the presence of a finite-width shear layer is not related to viscosity. The description in terms of a vortex sheet always breaks down even in ideal fluids when the system is examined at sufficiently small scales. Moreover, an effective interface of finite width between counter-propagating superflows naturally arises also in Gross-Pitaevskii simulations, where the vortex-array instability rates actually agree with those of the Kelvin-Helmholtz instability of a finite-width shear layer (see [44]).

I would thus generally refer to finite-width shear layer to introduce the ultraviolet cutoff, rather than to viscosity.

2) "Zwierlein’s group at MIT showed that a BEC subject to a synthetic magnetic field undergoes a snaking instability leading to a crystallization of the condensate in droplets separated by streets of quantized vortices."

I'm not sure to understand the relation between snaking and shear flow instabilities. In dynamical system theory, "snaking" typically refers to a transverse modulational instability that distorts a dark soliton stripe or a deep depletion region, eventually leading to the breakup into vortex filaments or droplet pairs. While the formation of a vortex array may occasionally result from a snaking instability, this typically has nothing to do with any subsequent instability of the array, the presence of shear flow, or shear layer instabilities such as the Kelvin-Helmholtz instability.

I would then ask the authors to clarify this issue or to remove the sentence.

3) The authors showed that introducing a finite core mass affects how the maximum instability growth rate depends on the number of vortices per unit length $N_v/L$​. Specifically, its scaling law shifts from a quadratic to a linear behavior for large $N_v$. This is a very interesting point that deserves further discussion. For massless vortices, the quadratic scaling of the most unstable mode arises because according to (1) the instability growth rate follows $\sigma \propto q N_v/L$ and the wavenumber of the most unstable mode follows $q^* \propto N_v/L$, leading to $\sigma^* \propto (N_v/L)^2$, which remains valid even in the limit of infinite vortices. Note that a generic unstable mode q would exhibit a linear scaling with $N_v/L$. For massive vortices $\sigma^*$ scales linearly with $N_v$. What about the other (unstable) modes?

It would be interesting to show a dispersion relation (growth rate vs $q$ for fixed N_v). Since the most unstable mode (and indeed all array modes) remains the same, I would expect a sublinear scaling for some of them.

4) A discussion of dissipative effects in the massless point vortex model, also with derivation of Eq. 20, have been already reported in [46]. I would suggest the authors to refer to this work at page 8.

5) In Sec. 5, the authors compare the predictions of their generalized point-vortex model with the recent results in [46]. This comparison provided an estimate for the mutual-friction coefficient $\alpha^{'}$around 0.75. This is an unusually high value with respect to what is commonly expected in the literature, although to my knowledge there are no direct measurements of this coefficient. While the coefficient here is basically phenomenological, it could be related to a number of phenomena, from the contribution of the Andreev-bound states and Iordanskii force to some unspecified technical effects. Have the authors any feeling about the physical origin of dominant effects here?

6) Pag 13:" ... while the mode q = 0 is always stable [i.e. σ(q = 0) = 0]."

If $\sigma$ here is the real part of the corresponding eigenvalue, then the mode is "marginally stable" and stability should be determined either numerically or going to second order in perturbations. I would tentatively suggest that ultimately, we might discover that it is unstable. Otherwise, I would expect that the system, when carefully prepared in the regular vortex necklace configuration, it would remain stable indefinitely. Is this the case?

7) Page 19: "... it is not possible to rigorously perform the linear-stability analysis for a (either massless or massive) vortex necklace subject to frictional forces. The reason is that, in the presence of dissipation, regular vortex necklaces (see Sec. 3.2 and Sec. 4.1) no longer constitute stationary solutions of the associated dynamical systems.."

I'm a bit puzzled about this point. It makes sense to perform stability analysis only on a solution of the system, either a fixed point or a limit cycle or a more complex attractor. If the regular vortex necklace isn't a solution in the presence of frictional forces, it implies that if we initialize the system in this configuration at t=0, the subsequent evolution won't be due to instability of the solution, but rather because this configuration isn't a solution of the system.

I concur with the authors that adapting Eqs. (16)-(18) to the vortex necklace by setting $a = 2 \pi r_0/ N_v $should give reasonable results, at least in some limits. However, from a conceptual standpoint, if the regular necklace is no longer a solution in the presence of these frictional forces, it implies that basically does not make sense to talk about instability of such a configuration. So either the instability no longer exists for circular geometries in the presence of frictional forces or that such forces should be modeled in a different way so to preserve the existence of the necklace solution.

I think that the authors should clarify this point.

Recommendation

Publish (easily meets expectations and criteria for this Journal; among top 50%)

  • validity: high
  • significance: high
  • originality: high
  • clarity: high
  • formatting: excellent
  • grammar: excellent

Author:  Andrea Richaud  on 2024-08-02  [id 4671]

(in reply to Report 1 on 2024-05-25)

We attach a pdf file with our detailed point-by-point reply, followed by a "diff" file highlighting the latest changes to our manuscript.

Attachment:

24_08_02_-_Reply_and_diff_HOWOMqR.pdf

---

## Round 1 · Referee Report · Anonymous (Referee 2) · 2024-6-5

Strengths

The manuscript is clearly written and enjoyable reading, the results are sound and the work contains a solid body of interesting results.

Weaknesses

The manuscript deploys/introduces a Newtonian mass term in the point-vortex Lagrangian formulation as a starting point. The goal is clearly to relate the predictions of the model to experiments but the origin or the form of the mass term is not clearly explained or justified.

Report

The manuscript is of high quality and well suited to SciPost Physics. It easily meets criteria and expectations for this journal. I have provided specific comments in an annotated .pdf for the benefit of the authors. The comments do not challenge the correctness of the derivations but point out that the applicability of the obtained results in the context of experiments may be affected by the inherent relationship between the vortex mass and the size of the vortex core.

Requested changes

Suggestions for optional changes can be found in the annotated .pdf.

Attachment

Recommendation

Publish (easily meets expectations and criteria for this Journal; among top 50%)

  • validity: high
  • significance: high
  • originality: high
  • clarity: top
  • formatting: excellent
  • grammar: excellent

Author:  Andrea Richaud  on 2024-08-02  [id 4670]

(in reply to Report 2 on 2024-06-05)
Category:
answer to question
reply to objection

We attach a pdf file with our detailed point-by-point reply, followed by a "diff" file highlighting the latest changes to our manuscript.

Attachment:

24_08_02_-_Reply_and_diff.pdf

---

## Round 2 · Referee Report · Anonymous (Referee 1) · 2024-8-3

Report

The authors have adequately addressed all the concerns raised in my report and have revised the manuscript accordingly. Therefore, I recommend its publication.

As a final minor suggestion, I would replace the word "only" with "mainly" in the new sentence at page 19:
"However, assuming that the breakdown of the necklace is only due to the Kelvin-Helmholtz mechanism...."

Recommendation

Publish (easily meets expectations and criteria for this Journal; among top 50%)

---

## Round 2 · Referee Report · Anonymous (Referee 2) · 2024-8-9

Report

The authors have carefully considered all feedback provided by both previous referees and I wish to recommend the manuscript to be published in SciPost in its present form.

Recommendation

Publish (easily meets expectations and criteria for this Journal; among top 50%)

---

## Round 2 · Author Response

Dear Editor,

thank you very much for sending us the reports of the two Referees. We are grateful to the Referees for their careful, positive and constructive reports that helped us improving our manuscript. We have revised the manuscript complying with the requests of both Referees, and we have replaced the arXiv manuscript with the revised one. Our detailed point-by-point response to the Referees has been uploaded.

In view of the already positive reports of both Referees and of the improvements that we have implemented in response to their comments, we are confident that this revised version of our paper can be accepted for publication in SciPost Physics.

Best regards,

The authors

---

## Round 2 · List of Changes

A detailed list of changes has been attached as a pdf file in the response to the Referees.

---

## Editorial Decision

published